# Challenges in Promoting Positive Youth Development through Sport

Carlos Ewerton Palheta [1,*], Vitor Ciampolini [1], Fernando Santos [2,3], Sergio José Ibáñez [4], Juarez Vieira Nascimento [1] and Michel Milistetd [1]

1   Sport Pedagogy Research Center (NuPPE), School of Sports, Federal University of Santa Catarina, Florianópolis 88040-900, Brazil
2   School of Higher Education, Polytechnic Institute of Porto, 4200-465 Porto, Portugal
3   Centre of Research and Innovation in Education inED, 4200-465 Porto, Portugal
4   Optimization of Training and Sports Performance Research Group (GOERD), Faculty of Sports Science, University of Extremadura, 10003 Cáceres, Spain
*   Correspondence: carlospalheta26@gmail.com

**Abstract:** Following previous calls regarding the importance of using sport as a platform for promoting youth psychosocial development and mental health, this case study aimed to understand the challenges faced by coaches and technical directors in the face of their individual efforts to infuse a positive youth development focus into a nationwide youth sports program in Brazil. After conducting semi-structured interviews, we conducted an inductive−deductive thematic analysis. High-order themes were organized according to the dimensions of the Framework for Planning Youth Sport Programs That Foster Psychosocial Development. The challenges faced appear to be interconnected. Since the program's goals are not evident to the stakeholders, parents seemed not to support the absence of participation in competitive events. The lack of a structured methodology for implementing the program hinders coaches from planning and delivering practices by concurrently integrating sport skills and life skills development. Furthermore, the program presents difficulties in ensuring its effectiveness through assessment processes.

**Keywords:** youth sport; life skills; mental health

## 1. Introduction

The World Health Organization defines mental health as a state of well-being that influences how one faces everyday life stresses and manages to be productive in their activities [1]. Discussions around mental health have been constant in recent years, mainly due to increased cases of disorders such as depression and anxiety in children and youth [2]. Accordingly, several organizations have created initiatives to promote mental health during childhood and youthhood. One example is the 2030 agenda of the United Nations, which is an intergovernmental effort establishing 17 goals to overcome the world's main challenges by 2030 and points to physical activity as a facilitative practice for mental health. Among the different types of physical activity, sport stands out for being a pleasant activity for young people [3], making it a promising and potentially engaging environment to support mental health [4]. Despite the lack of consensus on how to structure the sport environment to promote mental health [5], the Positive Youth Development (PYD) perspective may be a valuable path to be considered when structuring developmental programs [6–8].

Within sport, PYD is an asset-based approach that has supported coaches and programs in promoting the development of practitioners across various life domains [9]. One of the intended outcomes of PYD includes skills, values, and behaviors that can be learned and/or enhanced in one context and later transferred to other life domains, which are called life skills [10].

The concept of life skills has been adopted in youth-oriented endeavors by organizations such as the World Health Organization [11], the United Nations Children's Fund [12], and the United Nations Educational, Scientific and Cultural Organization [13]. In fact, the World Health Organization suggests that life skills such as effective communication, empathy, and emotional control contribute to socialization, healthy development, and mental health promotion [14]. However, fostering PYD and life skills through sport is a complex endeavor that requires understanding the purpose of a given sport program and the roles and responsibilities of parents, coaches, and other stakeholders [15]. Consequently, many researchers have developed frameworks to encapsulate the variables that influence sport-based youth programs with a PYD and life skill focus [8,16,17].

Petitpas et al. [16] proposed a four-dimensional framework for planning youth sport programs that foster psychosocial development. First, the authors acknowledged the need to understand the context (e.g., community-based program or competitive youth sport program) where the program is situated and explicitly define program objectives. Second, there is a need for coaches, technical directors, parents, and community members to support the implementation of program objectives (i.e., external assets). Third, the authors also raised the need to provide concrete opportunities for youth participants to learn and transfer a variety of life skills (i.e., internal assets). Finally, research and evaluation that accounts for the program's effectiveness are needed to help identify challenges and guide future decisions about program delivery.

Previous studies have used Petitpas et al.'s [16] framework to evaluate sport-based youth programs [8,18–20]. These studies highlighted the importance of creating deliberate efforts to teach life skills and creating a PYD climate. Conversely, they also pointed to the existence of multiple challenges such as the lack of funding and PYD objectives set by parents and coaches, youth's lack of motivation towards learning life skills, and the complex nature of evaluating life skills transfer [21,22].

As the contexts where the programs are situated highly influence the challenges they face [23], moving research beyond non-English-speaking countries is needed to increase our understanding of the challenges of infusing PYD into programming worldwide [8,24,25]. In Brazil, for example, sports programs are financed by the government across the country to allow youth to attain developmental outcomes [26]. Yet, most (if not all) programs in this context do not include an explicit PYD and life skill focus since research in this field is still in its infancy in Brazil [27,28], and the country does not have sport-specific public policies to promote mental health [29].

This study focused on a sport-based Brazilian afterschool program previously investigated by Palheta et al. [20], where the authors examined the alignment between program guidelines and delivery. Using Petitpas et al. [16], the authors found that program guidelines were aligned to PYD; yet, the guidelines were unable to provide clear guidance to coaches and technical directors on how to facilitate life skills development. Regardless of the lack of guidance, Palheta et al.'s [20] study also identified that coaches' and technical directors' individual efforts contributed to promoting PYD and sustaining the program's goals. When delivering and overseeing the program, coaches and technical directors promoted a positive environment (context), fostered meaningful relationships with youth participants and their parents (external assets), discussed life skills throughout the practices (internal assets), and attempted to observe life skills transfer (research and evaluation). Even though such findings illustrate some of the program's positive features, they also represent implicit and non-structured strategies, which may result in challenges that stakeholders will have to overcome [20].

From our understanding of the significant findings of Palheta et al.'s [20] preliminary study investigating this Brazilian nationwide program through Petitpas et al.'s [16] framework, we decided to conduct a second study focused on the challenges faced by coaches and technical directors. Identifying the challenges faced by coaches and technical directors of a Brazilian program will help researchers recognize the extent to which they are similar to the challenges faced by programs in other countries [21,22,25] and how to

overcome them. Therefore, this case study aimed to understand the challenges faced by sports coaches and technical directors in the face of their individual efforts to infuse a PYD focus into a nationwide youth sports program in Brazil. Two main questions guided this research: (I) What challenges do professionals in a sports program face as they individually strive to sustain a PYD focus? and (II) How can the PYD perspective leverage the individual efforts of coaches and technical directors in the program?

## 2. Materials and Methods

We used a case study and a qualitative approach to understand the participants' perceptions, specifically to provide an in-depth analysis of their challenges in delivering a sports program. From an ontological, epistemological, and axiological point of view, we framed our study in a constructivist paradigm [30]. Guided by constructivism, reality is socially constructed and context-specific, as it reflects particular interactions, experiences, and beliefs within a unique setting. Thus, we do not intend to generalize the results of this study but rather provide insights for researchers, coaches, and stakeholders from our interpretation of the data collected [31].

### 2.1. The Sport-Based Youth Program

In Brazil, sport is a physical activity recognized in the constitution across three different categories: (I) sport practiced freely without official rules and with the purpose of social integration and health promotion (i.e., participatory sport); (ii) sport practiced with official rules and with the purpose of competition (i.e., performance sport); and (iii) sport that prioritizes personal development and citizenship (i.e., educational sport). In the country, sport is recognized as a civic right conducive to developing skills, values, and behaviors that are important for living in society, and programs are fostered across the country to promote youth personal development [26]. The program examined in the present study is offered by a private organization responsible for implementing youth programs across Brazil. This private organization uses a range of activities, including dance, music, drama, and sport, to help youth learn life skills and contribute to society. The organization's mission is to promote quality of life through its services, and sustainability and innovation are among the values present in its culture. This program has been implemented since 2013 and was awarded internationally due to the civic and educational nature of its activities. Although this organization has several units across the country, reaching millions of young Brazilians, only one unit in the south of Brazil was examined in this study. This program is made available to children and youth ranging between 7 and 17 years old and involves two 50-min sessions per week as an afterschool program. It is worth mentioning that such a weekly length of sessions does not reach the minimum time recommended by the World Health Organization. Participants from 7 to 13 years old become involved in different sports, and participants from 14 to 17 years old practice a specific sport according to their interests. Participation in the program involves no costs, and there are no restrictions or criteria based on socioeconomic status. Additional information about the organization and its activitiesremains confidential for ethical reasons.

### 2.2. Participants

As this study builds on Palheta et al.'s [20] study, the same participants were purposively selected. First, the two technical directors (National Department and State Department) interviewed in Palheta et al.'s study were intentionally selected considering their important role in elaborating and overseeing the program. The other participants (i.e., Technical Directors of Local Units and Coaches) were selected through snowball sampling [32]. Participants were six technical directors and five youth sport coaches who have a bachelor's in physical education and were paid to work in the program (see Table 1 for further demographic information). All the participants, including technical directors, have previous experiences as youth sport coaches and were actively involved in structuring and overseeing (i.e., technical directors) or delivering (i.e., coaches) the program for a

minimum of one year. Technical directors were involved in diverse levels of management (nationwide, statewide, and local) and were responsible for providing the guidelines for sport-based youth programs offered by their organization across the country (see more information regarding the participants in Palheta et al. [20]). We created pseudonyms to protect participants' anonymity.

**Table 1.** Participants profile.

| | Participants | Age | Gender | Management Level | Experience in the Program (years) |
|---|---|---|---|---|---|
| **Technical Directors** | Mary | 52 | Female | National Department | 24 |
| | Sarah | 43 | Female | State Department | 24 |
| | Adam | 55 | Male | Local Unit 1 | 5 |
| | Ana | 34 | Female | Local Unit 2 | 7 |
| | David | 28 | Male | Local Unit 3 | 7 |
| | Samantha | 32 | Female | Local Unit 4 | 11 |
| **Coaches** | Albert | 28 | Male | Local Unit 1 | 3 |
| | Susan | 26 | Female | Local Unit 1 | 1 |
| | Bernard | 40 | Male | Local Unit 2 | 7 |
| | Gilbert | 36 | Male | Local Unit 3 | 5 |
| | Henry | 30 | Male | Local Unit 4 | 5 |

*2.3. Data Collection*

Ethical clearance was received from the first author's university research and ethics board. The organization investigated also authorized the research, and the technical directors and coaches were contacted. The interview took place on the dates and times suggested by the participants. To foster an open discussion about program delivery and minimize social desirability effects, the research team emphasized that the study's objective was not to evaluate the participants' performance and served the purpose of promoting learning between researchers and stakeholders across the globe.

The first author conducted semi-structured interviews with the participants. The interview guide was organized into five sections. The first section involved questions about the participants, such as age, involvement in the program, professional training, and previous experiences. The four later sections were based on the dimensions proposed by Peptipas et al.'s [16] framework and involved questions about possible challenges faced in program delivery and oversight. Examples of questions include: How do you promote an environment without pressure for results and performance? Do you face any challenges (e.g., context)? Do you foster relationships with youth participants' parents? Is it challenging (e.g., external assets)? How do you teach life skills in your practices? Do you face any challenges in doing this (e.g., internal assets)? How do you assess the development of the participants? Do you face any challenges in conducting your assessment practices (e.g., research and evaluation)? Before each interview, PYD and life skills were defined for the participants. PYD was defined as an asset-based approach that involves strategies to promote youth participants' personal development. Life skills were defined as values, attitudes, and positive behaviors important in sport and other life domains. Interviews lasted a mean of 45 min, and the first author transcribed the interviews into a 74 single-spaced page document.

### 2.4. Data Analysis

We conducted a thematic analysis [33] to identify patterns of meaning in the participants' perceptions. This analysis involved six steps that were interactively and flexibly followed: (a) familiarization with the dataset; (b) creating codes; (c) organizing codes into initial themes; (d) reviewing and refining themes and naming themes and subthemes; and (e) producing the final report. After uploading the material into Nvivo 9.0, the first author immersed himself in the raw data by analytically reading the document multiple times, taking reflective notes, and creating initial codes that evoked the data around the research question. Second, the deductive component of the analysis was conducted through the four dimensions of Petitpas et al.'s [16] framework, in which the initial codes were organized and converted into these high-order themes. Then, the first author engaged in rereading these initial themes to inductively identify data patterns and develop the final list of themes. Finally, a report including quotes and descriptions that best represented each theme was prepared.

### 2.5. Rigor

Thematic analysis emphasizes the researcher's inevitable subjectivity and active role in coding and generating themes, which requires deep involvement and reflection on the data, as well as respect for the values of the adopted paradigm [34]. Therefore, considering the relativist ontology of the constructivist paradigm [35], we adopted several procedures to increase rigor according to the nature of the present study. The coauthors served as 'critical friends' to the first author by questioning his interpretations and providing alternative insights about the analysis. Throughout several months of team meetings and discussions around the analysis, the first author was able to go back and forth multiple times in the analysis to find a "good fit" and a compelling story to represent the dataset. When the authors deemed to have a data analysis encompassing internal homogeneity and external heterogeneity among the themes and subthemes, they prepared a final report that turned into the results section. The first author also kept a reflexive journal and participated in a research group where the guidelines provided by Braun et al. [33] concerning thematic analysis were constantly discussed. It is worth mentioning that none of the researchers involved in the study had direct involvement with overseeing or delivering the program.

## 3. Results

Table 2 presents the four deductive high-order themes according to the dimensions of Petipas et al.'s [16] framework and several subthemes. Each high-order theme contains the participants' objectives for the program, the challenges they face, and their respective causes.

**Table 2.** Final set of themes and subthemes.

| Themes | Subthemes | | |
| --- | --- | --- | --- |
| | Participants' objectives | Challenges | Causes |
| Context | Foster an environment where there is no pressure to attain performance outcomes and results | Develop a coherent program | Competition as the priority |
| | | | Comparisons to elite youth sport clubs |
| External assets | Establishing positive relationships with parents and youth participants | Align parents' objectives with youth participants' needs and the program's guiding philosophy | Parents that only value performance outcomes |
| | | Involve all youth participants in life skill development | Lack of motivation toward life skills showed by adolescents between 14 and 17 years old |

**Table 2.** *Cont.*

| Themes | | Subthemes | |
| --- | --- | --- | --- |
| Internal assets | Fostering life skills | Align the program's pedagogy with life skills development | No guidelines for coaches to foster life skills |
| | | Plan training sessions with a life skill focus | Few discussions between stakeholders |
| | | | High number of youth participants per session |
| | | Teaching life skills | Short duration of practices |
| | | | High number of youth participants per session |
| Research and evaluation | Assuring the program's effectiveness | Assess program outcomes | No knowledge about evaluation processes and tools |

*3.1. Context*

Both coaches and technical directors recognized the importance of creating a psychologically safe environment for youth participants. They also stressed the need to avoid negative sport experiences that may derive from engaging in performance-focused climates. In fact, participants seemed to believe that avoiding negative experiences and competition could be enough to foster positive sport experiences and, in turn, life skill development: "Competition does not involve all [youth participants] and overvalues the most skilled. We avoid that. We do not rank the best or the worst [youth participants] . . . . The one who was first, second or third" (Sarah—Technical director from the State Department). Thus, the program eliminated competitive participation and any sport performance focus to foster PYD outcomes: "Our stand is that sport is a platform for personal development. This [approach] demands they [youth participants] have positive and meaningful experiences that enable reflection and self-awareness without competitive results pressuring them" (Mary—Technical director from the National Department). However, participants found it challenging to explain this approach to the community. This was the case mainly due to the fact that no youth participants were playing in competitive events: "We do not participate in competitions, and people ask us why. It is hard for them [community] to understand that what we focus on [life skills development] is not dependent on competitive results" (Coach Gilbert).

The main challenge was dealing with a context where competition was deemed necessary and highly overvalued. Such culture and the social forces in place at a community level made parents reluctant to support the program's mandate. Since many community members learned from the competition and lived in a highly competitive environment, there was the belief that participating in competitive events should be compulsory and a key component of any program: "Adults want youth to compete, to win games and championships. For them, sport is just that" (Ana—Technical director from the Local Unit). Since competitive youth clubs were massified across Southern Brazil, there were high expectations for sport-based youth programs in the same context to share a common philosophical focus on winning and performance, which created additional challenges for the participants:

"Here in [name of the city], we have [name of a club], which is a well-renowned basketball club. We also have a futsal club. So, many believe we will achieve the same level of success these clubs have. We face the challenge of convincing others that our pedagogical approach is different" (David—Technical director from the Local Unit).

*3.2. External Assets*

Participants deemed it crucial to foster positive relationships with parents and youth participants. Indeed, coaches recognized that quality relationships with youth participants

created solid grounds for fostering PYD through sport: "To teach life skills, I need to foster their [youth participants'] trust" (Coach Bernard). Furthermore, both coaches and parents attempted to convince parents to buy in and collaborate to promote PYD and forge partnerships with them: "Of course, parents are key in that process [PYD]. Who does not want their child to learn positive behaviors? Our program has promoted events where parents can attend and understand our methodology, they [parents] need to support our cause" (Mary—Technical director from the National Department).

Nevertheless, participants identified two main challenges while trying to engage parents and teach youth participants some life skills. Concerning the first challenge, parents' objectives were not aligned with youth participants' needs and the program's guiding philosophy. Although the program's mission and objectives were explained and discussed with parents, they still showed no interest and were strongly concerned about sport skill development and performance. It was deemed difficult to reconcile such differences and attain a sense of redundancy, where parents, technical directors, and coaches focused on the same outcomes: "Over time, they [parents] continue to expect a training session that just focuses on technical skills" (Ana—Technical director from the Local Unit). Conversely, coaches emphasized the notion that youth participants need a fun environment where they can connect with their peers, which contradicts parents' objectives:

"With them [youth participants], I do not have this challenge. What they want is to have fun with their colleagues. If the activity is fun and engaging, they are not reluctant to participate" (Coach Gilbert).

Parents who had younger children (i.e., up to 13 years old) were even more aware of the urgent need to focus on sport skill development and performance to create chances for a professional career in sports: "Generally, parents who have younger children wish they become great football, basketball, or futsal players" (Samantha—Technical director from the Local Unit).

Another challenge identified by the participants was to motivate youth toward life skills development. There was an evident lack of motivation toward learning life skills displayed by adolescents between 14 and 17 years of age. Conversely, younger youth participants were open to learning life skills through sport: "I feel the older ones [youth between 14 and 17 years old] are not receptive when I discuss behaviors and values they do not always engage in our discussions" (Coach Henry).

*3.3. Internal Assets*

The program fosters the teaching of life skills during the activities offered to promote personal development. Coaches play a central role in this mission as the technical directors provide full autonomy for them to plan and deliver practices:

"We have the autonomy to plan. Until now, technical directors never inhibited my intervention efforts. They [technical directors] trust coaches' work, and we need to consider the program's philosophy. We work with children and youth, and our role as coaches is to teach values and behaviors" (Coach Albert).

However, several challenges were identified. Technical directors struggled to define a sound approach to teaching life skills that could guide coaches' practices and, in turn, foster better PYD outcomes: "We have tried to use several strategies to educate coaches on how to foster that [life skills], but as of today we do not have a specific methodology [to attain this purpose]" (Mary—Technical director from the National Department). Thus, no guidance was provided to coaches through guiding manuals and other resources, which was deemed a significant challenge. The program's effectiveness relied on coaches' awareness of youth participants' developmental needs and knowledge of PYD: "We don't have a manual that helps coaches teach life skills. Each coach has the freedom to define their own strategies" (Sarah—Technical director from the State Department).

Considering coaches' autonomy to plan and implement a PYD mandate, several challenges must be considered. First, the lack of time spent (i.e., only two sessions per week) with youth participants did not allow coaches to consider this: "We meet a few

times per week, which makes it hard to understand everyone's personality. Ideally, we would have more time because I believe it is important to know them [youth] better to understand which life skills to teach" (Coach Henry). Second, coaches indicated they explicitly discussed life skills in group meetings and reflections promoted in practice. However, the sessions were deemed too short to infuse a life skill focus systematically and concurrently teach sport skills: "The main way to work on those aspects [life skills] is through group discussions and if needed, individual conversations. Sessions last for 50 min which makes it a challenge to do this [use these strategies] on every practice" (Coach Bernard). Finally, coaches claimed the high number of youth participants per team (i.e., above 25) created challenges for coaches to foster PYD outcomes. Even though coaches were satisfied to have so many youth participants interested in joining the program, they claimed it created quite a challenge to establish bonds and discuss life skills with them during practices: "The number of youth participants per team is high. On one hand, that is great. The program needs youth to function but that also creates barriers for teaching life skills" (Coach Susan).

*3.4. Research and Evaluation*

Participants alluded to the notion that evaluating program outcomes was quite challenging. Such perception may have derived from the lack of awareness about how to infuse PYD and a life skill focus into programming and to understand an operational definition of PYD that reflected the program's guidelines: "It is very hard to assess these aspects [life skills] because it involves emotional, affective, and educational [aspects].I have no idea how to do that [evaluation]" (Samantha—Technical director from the Local unit). In tandem, participants did not know any assessment tools and showed a limited notion about what they wanted to assess concerning program outcomes:

"Today, we have tools to assess psychomotor variables, but not for life skills development. We know what we want [for the program], but we still have some issues to change . . . it is still not clear which outcomes [life skills] we have achieved" (Mary—Technical director from the National Department).

Moreover, coaches were unaware of any measures that could help them self-evaluate and understand the extent to which they were teaching life skills more systematically. Thus, coaches used their perceptions to assess program delivery and youth participants' outcomes, which were influenced by their narrow understanding of PYD and life skills: "Many youth participants improve their behavior throughout the program. This is quite evident when I observe older students and see they are more polite and respectful nowadays but it is hard to monitor [these outcomes] and make sure I was responsible for that" (Coach Albert).

## 4. Discussion

Following previous calls regarding the importance of using sport as a platform for promoting youth psychosocial development and mental health, this case study aimed to understand the challenges faced by sports coaches and technical directors in the face of their individual efforts to infuse a PYD focus into a nationwide youth sports program in Brazil. As in previous studies [21,22], the framework proposed by Petitpas et al. [16] allowed us to identify the challenges faced in different program dimensions. In general, the challenges appear interconnected and triggered by some decisions made and the lack of guidance in the context dimension. We discuss below how this dimension connection influenced the challenges faced by the participants.

In the context dimension, the program seeks to provide a sporting experience where youth are exempt from competitive constraints. Many sport-based youth programs aligned with a PYD approach abandon sports performance goals due to the belief that focusing on performance negatively influences personal development [36]. In an attempt to secure youth from the negative experiences sport can promote [37], the program eliminates participation in any form of competition, which leads to an enormous challenge when explaining this decision to parents and the community. Overcoming this challenge requires

recognizing that, although favorable, engagement in the sport context is not the only form that can promote PYD [38]. All the features of sport (e.g., rules, decision making, positive and negative experiences, relationships, and competition) are responsible for making it a paramount platform for PYD [6]. Therefore, allowing youth to experience victory and defeat in sport can even favor life skill development [39,40]. Additionally, competition may help coaches visualize signs of the near transfer of life skills [15], for example, leading the team against a good opponent and regulating their emotions in a late-game situation, thus identifying signs of life skill transfer.

One of the program's goals was to forge meaningful relationships with parents and youth [20], which aligns with previous recommendations for quality PYD programming [16,41]. However, according to the participants, parents' ultimate concerns with the development of sport skills cause one of the challenges faced in the external assets dimension. The development of technical and tactical skills is imperative for any sport program, regardless of being based on PYD or any other developmental focus [42]. Thus, studies recommend programs to incorporate goals related to PYD and sport performance concurrently [25,36]. It seems that the challenges found in the context dimension led to another challenge in the external asset dimension. In other words, recognizing PYD through sport as a dichotomy to sport skills development and competition leads to some challenges with parents who expect their children's competitive involvement and development. Overcoming this challenge is fundamental, as parents are external assets co-responsible for promoting PYD, in addition to being another source of information in terms of life skill transfer to the family environment [15,16,43]. An alternative is to create channels of communication between the program and parents that effectively clarify PYD goals and support life skill development.

In the internal assets dimension, the challenges identified seemed to be consequences of the program's lack of support and clear instructions. Coaches reported the lack of time to implement life skills strategies and the resistance presented by youth—especially the older ones between 14 and 17 years of age—when organizing group discussions around life skills. Even though we acknowledge coaches' efforts to plan and teach life skills in their practices, their mindset still appeared to present the same dichotomy as youth participants' parents. Indeed, coaches seem to perceive life skill activities apart from sport skill activities. Therefore, organizing coach education courses around PYD and life skill development and providing clear directions for planning and delivering sport practices is crucial for aligning the program objectives with coaches' behaviors on the ground [44,45]. These programs should demonstrate, for example, how to combine the coaching of technical, tactical, and physical aspects of sports with concrete life skill development strategies.

Even though youth participants were not interviewed in this study, research conducted in the Brazilian context has previously pointed out resistance to discussing life skills [46]. In Ciampolini's et al. [46] study, youth between the ages of 13 and 15 who had never participated in sport programs that included an explicit approach towards teaching life skills initially perceived it as an odd approach. Although group discussion is a strategy indicated by scholars to promote life skill development [41,47], youth may find it more motivating when life skills are integrated with the development of sport skills [48]. Moreover, the strategies used to foster life skills within sport-based youth programming may need to be contextualized to the sport and age group of the participants [10,17]. Furthermore, the lack of relevancy provided to PYD and life skills in Brazil [28] may also influence how explicit approaches are considered by youth participants and a vast array of sport stakeholders.

Concerning research and evaluation, the program faces a common challenge to PYD programs, which is to measure the impact and effectiveness of the activities organized. Since PYD and life skill transfer are complex processes, selecting procedures and instruments that encompass the bidirectional nature of human development is a challenging endeavor [15,49–51]. Such a challenge exists even in English-speaking countries where the PYD approach has been widespread since the 1990s [51]. In Brazil, access to quantitative methods is even more scarce, considering the lack of instruments translated and validated

to the Portuguese language [52–54]. Thus, contextualized research and evaluation protocols that consider cultural nuances are needed to move the field forward and increase our ability to understand program delivery and outcomes across non-English-speaking countries [25,50].

## 5. Conclusions

Through the lens of PYD, we identified that the challenges faced by participants are interconnected across the framework proposed by Petitpas et al. [16]. In other words, since the program's goals are not clarified to the stakeholders (i.e., context challenge), parents seem not to support the absence of participation in competitive events (i.e., external asset challenge). Moreover, the lack of a structured methodology for implementing the program hinders coaches from planning and delivering practices by integrating sport skill and life skill development concurrently (e.g., internal assets challenge). As a result of the challenges faced in the other dimensions, the program presents difficulties in ensuring its effectiveness through assessment processes (e.g., research and evaluation challenges).

To overcome such challenges, we suggest the programs develop means of communication with parents, youth, and the community to clarify their ultimate goal (i.e., promoting personal development). Moreover, instead of excluding competitive events and performance goals from their guidelines, integrating sport skill with life skill teaching and organizing competitive events in a developmental manner is a possible way to meet parents' expectations and engage youth in life skill activities. With the alignment between the context and the external assets dimension, the creation of guidelines for teaching life skills through sport may be facilitated. Considering that these actions must precede the search for developmental outcomes, we suggest the program create self-assessment mechanisms that allow coaches to reflect on their practices and obtain preliminary results on developmental outcomes.

Among the limitations of this study, the different levels of involvement of the participants with the program (from 1 to 24 years of experience) may have influenced how they declared their perceptions. It is also worth mentioning that the perceptions of participants and their parents were only considered indirectly through the account of technical directors and coaches. Moreover, the lack of direct contact with participants and their parents hindered us from identifying their perceptions regarding mental health promotion and life skill development as an outcome of involvement in the sport program. Despite the selected participants' perceptions aligning with the objectives of this study, including the perceptions of parents and youth, conducting systematic observations could have contributed to a better understanding of the program and may represent future steps for research in the field. Thus, integrating a variety of data collecting methods, including systematic observation methods, may take research in the field a step further and increase researchers' ability to tap into the cultural nuances and particularities of program delivery in less explored socio-cultural contexts such as Brazil.

**Author Contributions:** Conceptualization, C.E.P., V.C., F.S. and M.M.; data curation, C.E.P.; investigation, C.E.P. and M.M.; methodology, C.E.P., V.C. and F.S.; project administration, J.V.N. and M.M.; supervision, J.V.N. and M.M.; validation, V.C., S.J.I., J.V.N. and M.M.; visualization, J.V.N. and M.M.; writing—original draft, C.E.P., V.C., F.S., S.J.I. and M.M.; writing—review and editing, C.E.P., V.C., F.S. and S.J.I. All authors have read and agreed to the published version of the manuscript.

**Funding:** This research received no external funding.

**Institutional Review Board Statement:** This research was approved by the Ethics Committee for Research on Human Beings (CEPSH) at the Federal University of Santa Catarina (CAAE 74156517.0.0000.0121), following the research standards required by Resolution 466/2012 of the National Health Council of Brazil.

**Informed Consent Statement:** The research participants signed a consent form after ethical approval at the first author's university. The information obtained in the data collection was used only for academic purposes, and the anonymity of the research participants was maintained.

**Data Availability Statement:** Not applicable.

**Acknowledgments:** Coordination for the Improvement of Higher Education Personnel—Brazil (Coordenação e Aperfeiçoamento de Pessoal de Nível Superior—CAPES).

**Conflicts of Interest:** The authors declare no conflict of interest.

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
