# Peer review of "Challenges in Promoting Positive Youth Development through Sport"

_sustainability, doi:10.3390/su141912316_

Round 1
Reviewer 1 Report
I have to congratulate you for the work done; despite the fact i still have 2 big concerns. From my point of view, the work needs two improvements focused on the analysis:
2.5 Rigour 173-186
- Objective reliability with some kind of statistical test. For sure following Hernández-Mendo et al 2014 you could get a proper data quality
Hernández Mendo, A., Castellano, J., Camerino Foguet, O., Jonsson, G. K., Blanco Villaseñor, Á., Lopes, A., & Anguera Argilaga, M. T. (2014). Programas informáticos de registro, control de calidad del dato, y análisis de datos. Revista de Psicologia del Deporte, 2014, vol. 23, num. 1, p. 111-121.
Results 187
- Results with some statistic analysis. There is a wide paper examples with interviews associate to any type of data analysis (CrossTab , for instance)
Please, I beg you to carry out the two indications; otherwise I can not accept. I consider it is a very good job, when these two requirements are fulfilled.
Author Response
ANSWER:
Thank you for your attention with our work.
Although we understand the importance of statistical tests for some research (including qualitative research), this type of procedure does not fit with the paradigm we adopted in our study. Following the seminal work of Egon Guba, Yvonna Lincoln e Norman Denzin (Denzin, Lincoln, 2018), we chose one of the five main paradigms used in research, namely constructivism. As we stated in section 2, a constructivist ontology, epistemology, and axiology represent understanding that “the truth” is relative and socially constructed by one’s subjective experiences, values, and beliefs. That is why we used thematic analysis (Braun, Clarke, and Weate, 2016), as this qualitative analysis method take into account the researcher's inevitable subjectivity and active role in finding patterns across the data set, which requires deep involvement and reflection on the data. Therefore, even though we acknowledge the reviewer’s concern with the trustworthiness and quality of our analysis, we highlight that the procedures and techniques used should be aligned with the qualitative approach we adopted (Smith and McGannon, 2018).
In order to present such arguments to the reader, we further clarified our ontological, epistemological, and axiological position in items 2 and 2.5.
Below, we present the full reference of the citations mentioned that support our statements:
Braun, V., Clarke, V., & Weate, P. (2016). Using thematic analysis in sport and exercise research. In B. Smith & A. C. Sparkes (Eds.), Routledge handbook of qualitative research in sport and exercise (pp. 290-309). Routledge.
Denzin, N. K., & Lincoln, Y. S. (Eds.). (2018). The SAGE handbook of qualitative research (5 ed.). SAGE.
Smith, B., & McGannon, K. R. (2018). Developing rigor in qualitative research: Problems and opportunities within sport and exercise psychology. International Review of Sport and Exercise Psychology, 11(1), 101-121. https://doi.org/10.1080/1750984X.2017.1317357
Reviewer 2 Report
Dear Authors,
The topic is relevant and fits the purpose of the journal.
The abstract of the article meets the requirements.
However, the main changes to this paper should be made in the theoretical part of this paper (see the full introductory part).
The title draws the main attention to the construct of positive youth development, but the theoretical part discusses both mental health and life skills, psychosocial development. In this section, I would suggest focusing on PYD, emphasizing the difficulties experienced in promoting it.
Research methods and results are presented correctly.
Part of the discussion is well written, the conclusions are presented clearly.
Author Response
ANSWER:
Thank you for your comment.
Positive Youth Development (PYD) is a broad perspective frequently adopted in the research agenda to refer to concepts and approaches that aim to contribute to the psychological and social development of young people for their later success in life. By adopting the concept of life skills and the mental health construct, we are referring to some of the desirable psychosocial outcomes of PYD. That is why we used Petitpas et al.’s (2005) framework on PYD and cited studies in our introduction that place life skills and mental health under the “PYD umbrella”.
When approaching the construct of mental health, we cited documents from recognized entities (e.g., World Health Organization) that declare the importance of developing life skills to contribute to children and youth positive development.
However, to make our arguments clearer to the reader, we added a sentence in the first paragraph and highlighted the two main questions that guided our research.
Reviewer 3 Report
The content of the article is within the research area of ​​Special Issue "Sustainable Physical Activity and Student's Health".
The presented constructivism as a research paradigm with an idiosyncratic construction and subjective nature (line 105-106) is unclear as a research method due to the lack of definition of how and what interactions between participants were studied.
Research questions and hypotheses were not clearly formulated.
It is not known what the criterion for selecting the respondents was (from how many of the possible respondents were six technical directors and five youth sport coaches?), How the selection of the research sample was carried out (random, purposeful?) And when exactly the interviews were conducted.
What exactly was the implementation of the program about? Merely stating that two 50-minute sessions per week as an afterschool program, as well as "more information regarding 125 the program can be found in Palheta et al. Study" (line 123-126), is not enough for the reader to understand the description of the presented research results or re-conduct them - in accordance with the stated goal: "... the research team emphasized that the study's objective was not to evaluate 144 the participants' performance and served the purpose of promoting learning between 145 researchers and stakeholders across the globe" (s . 145-146). The research tool was also not described in detail. How many and what questions did the interview questionnaire consist of? The information that they can be made available upon request (line 149) also does not facilitate further understanding of the discussed results of own research.
Therefore, it should be attached to the text so that other researchers (since the purpose of the publication is to promote such research in the world) could attempt to undertake it in their countries and then compare the results.
The discussion indicated the lack of involvement in the parents' program due to their expectations - that is, the participation of their children in sports when there is competition, not play only. It is worth noting in the conclusions that it is important to be aware of all program participants that by participating in sports activities it is possible to learn life skills that children and young people acquire and use in non-sports life. Both those proposed by WHO: communication and interpersonal skills, decision making and critical thinking, self-management (Skills for health: skills-based health education including life skills: an important component of a child-friendly / health-promoting school) and WHO / UNICEF: interpersonal, building awareness, building your own value system, decision making, coping and managing stress (Life skills training program).
Conclusion. The article is acceptable for publication, after significant methodological additions and discussion of activities undertaken within the PYD program - as above.
Author Response
ANSWER:
Thank you for your comments.
In order to align our methods with our paradigmatic positionality, we collected participants’ unique perceptions through semi-structured interviews, analyzed data through thematic analysis, and adopted ‘critical friends’ to promote rigour to our interpretation. Following the comment made by the reviewer, we reframed items 2 and 2.5 to better clarify the connection between the paradigma we adopted and our qualitative methods.
The research questions were added to the end of the introduction.
The participant selection criteria were further clarified in item 2.2.
More information about the program was added in item 2.1. However, it should be noted that other information about the organization and its activities will remain confidential for ethical reasons, respecting the request of the technical directors. This information was also described in item 2.1.
We provided more information about the interview guide in item 2.3. The number of questions and the description of all of them cannot be done, as we used a semi-structured guide where additional questions were asked through a dynamic dialogue established between the researcher and the participants.
Reviewer 4 Report
Thank you for giving me the opportunity to review this very interesting manuscript entitled “Challenges in Promoting Positive Youth Development through Sport », which seeks to understand the challenges faced by coaches and technical directors in their individual efforts to infuse a positive youth development focus into a national youth sports program in Brazil.
The topic is relevant, the manuscript is well written, really interesting and this case study can contribute to the extant literature providing new theoretical insights, avenues for research and will interest readers.
I would like to make some suggestions that should be considered by the authors in order, in my opinion, to improve the quality of the manuscript.
Major concerns
More information should be useful concerning the interviews:
-Arrival and introductions, introducing the research, conducing and ending the interview (and durations).
The different level of experience in the program (from 1 to 24 years) of the interviewed technical and Coaches should be considered.
I also suggest to adding other relevant references:
Arbesman, M., Bazyk, S., & Nochajski, S. M. (2013). Systematic review of occupational therapy and mental health promotion, prevention, and intervention for children and youth. The American journal of occupational therapy : official publication of the American Occupational Therapy Association, 67(6), e120–e130.
Mansfield, L., Kay, T., Anokye, N. et al. A qualitative investigation of the role of sport coaches in designing and delivering a complex community sport intervention for increasing physical activity and improving health. BMC Public Health 18, 1196 (2018). https://doi.org/10.1186/s12889-018-6089-y
O'Connor, C. A., Dyson, J., Cowdell, F., & Watson, R. (2018). Do universal school-based mental health promotion programmes improve the mental health and emotional wellbeing of young people? A literature review. Journal of clinical nursing, 27(3-4), e412–e426. https://doi.org/10.1111/jocn.14078
Minor
Page 1
Abstract
I suggest not to using reference in the text (line 20)
Introduction
Line 33-34: I suggest to adding examples for better clarity and comprehension.
Second paragraph: I suggest to adding a definition of « sport ». (I’m not a native english speaker but a French researcher. In France, « sport » only concerns practicing in a sport federation (e.g., Federation Française de Football). Practitioners are registered in an approved club, pay for a federal sports license including insurance and have the opportunity to participate in official competitions in a sporting discipline). If it is the same in Brazil please add precision.
Page 2
Lines 47-51: I suggest to adding more recent references (e.g., Arbesman et al., 2013; Chaudhary & Mehta 2012; Nasser & Rajibi, 2010; Savoji et al., 2013; O’Connor et al., 2018 or Sobhi-Gharamaleki et al., 2010)
Arbesman, M., Bazyk, S., & Nochajski, S. M. (2013). Systematic review of occupational therapy and mental health promotion, prevention, and intervention for children and youth. The American journal of occupational therapy : official publication of the American Occupational Therapy Association, 67(6), e120–e130.
O'Connor, C. A., Dyson, J., Cowdell, F., & Watson, R. (2018). Do universal school-based mental health promotion programmes improve the mental health and emotional wellbeing of young people? A literature review. Journal of clinical nursing, 27(3-4), e412–e426. https://doi.org/10.1111/jocn.14078
Line 58: I suggest to replace « Petitpas et al. [1] » by « the authors » to avoid repetition.
Page 3
Line 100: replace « counties » by « countries »
Line 117: please add examples concerning the “activities” for better clarity
Line 123-126: I suggest to adding information concerning the “WHO” recommendation concerning physical activity in children (Does the program achieve 1 hour of physical activity per day?) Page 4 Generally coding is performed by 2 authors (e.g., Mansfield et al., 2018) please justify. Please add information concerning interviews duration and datat recording duration. How the sample was selected? Please add precision. Page 6 Context part of the manuscript: I suggest using italics for the transcriptions of the technical’ and coaches’ dialogues, for better clarity and reading.
Limitation:
In view of the research work of Smith and Sparke (2016; “Interviews: Qualitative interviewing in the sport and exercise sciences ») other limits could be evoked.
References:
Please check for the instructions for authors: (e.g., references form 2, 3, 25, 26, 28, 29, 30, 33, 35, 36, 37, 39, 40, 41, 42, 43, 44, 45, 46-54).
Author Response
ANSWER:
We appreciate your considerations.
More information on the interviews was provided in item 2.3 and on the selection of participants (item 2.2).
Regarding the different levels of experience of the participants in the program, we included this information as a limitation of the study in the last paragraph of the conclusion.
A definition of sport in the Brazilian context was provided in item 2.1.
We describe the types of activities offered by the organization in item 2.1.
In response to your suggestion, we describe an example of a government initiative to promote mental health in paragraphs 1 and 2 of the introduction.
Although we understand your suggestion to replace the references in paragraph 4 of the introduction, we understand it is important to use the references of documents from important organizations such as WHO, UNICEF, UNESCO, as our objective is to present how these organizations approach the concept of life skills. Yet, we did use some of the references you suggested in other sections of the manuscript.
We added the WHO recommendations on physical activity for children and young people in item 2.1, where we relate this to the time of practice and the weekly length of the program.
Minor revisions have been made both in the text and in the references.
Round 2
Reviewer 1 Report
.
Reviewer 3 Report
Thank you for introducing corrections and research questions. I hope that other researchers will take up a similar topic and contact the authors to clarify any ambiguities about the methods used with our paradigmatic positionality.